# An application of COM-b model to explore factors influencing veterinarians' antimicrobial prescription behaviors: Findings from a qualitative study in Bangladesh

Shahanaj Shano[1,2], Md Abul Kalam[2]*, Sharmin Afrose[3], Md. Sahidur Rahman[4], Samira Akter[5], Md Nasir Uddin[6], Faruk Ahmed Jalal[7], Pronesh Dutta[1], Mithila Ahmed[8], Khnd Md Mostafa Kamal[9], Mohammad Mahmudul Hassan[10,11], Maya L. Nadimpalli[12]

1 Institute of Epidemiology Disease Control and Research (IEDCR), Mohakhali, Dhaka, Bangladesh, 2 Global Health and Development Program, Laney Graduate School, Emory University, Atlanta, Georgia, United States of America, 3 Department of Anthropology, University of Manitoba, Winnipeg, Manitoba, Canada, 4 Bangladesh Country Office, Eastern Mediterranean Public Health Network (EMPHNET), Dhaka, Bangladesh, 5 Department of Anthropology, Jahangirnagar University, Savar, Dhaka, Bangladesh, 6 International Center for Diarrheal Disease Research, Bangladesh, Mohakhali, Dhaka, Bangladesh, 7 Humanity and Inclusion, Bangladesh Country Office, Dhaka, Bangladesh, 8 National University, Gazipur, Bangladesh, 9 Deakin Business School, Deakin University, Melbourne, VIC, Australia, 10 Queensland Alliance for One Health Sciences, School of Veterinary Science, The University of Queensland, Brisbane, QLD, Australia, 11 Faculty of Veterinary Medicine, Chattogram Veterinary and Animal Sciences University, Khulshi, Chattogram, Bangladesh, 12 Gangarosa Department of Environmental Health, Rollins School of Public Health, Emory University, Atlanta, Georgia, United States of America

* mkalam@emory.edu

**Data Availability Statement:** Dataset is not publicly shareable because the participants of the

## Abstract

The integration of behavioral theories in designing antimicrobial stewardship (AMS) interventions aimed at optimizing the antimicrobial prescription in veterinary practice is highly recommended. However, little is known about the factors that influence veterinarians' antimicrobial behavior for food-producing animals in lower- and middle-income settings like Bangladesh. There is a large body of research on the factors that influence veterinarian behavior of prescribing antimicrobials, however, there is a need for more studies that use comprehensive behavior change models to develop and evaluate interventions. Applying the Capability, Opportunity, and Motivation for Behavior (COM-B) model, this qualitative study attempted to address this gap by conducting 32 one-on-one semi-structured interviews with registered veterinarians in Bangladesh. In alignment with COM-B constructs and the theoretical domain framework (TDF), thematic analysis (both inductive and deductive inferences) was performed to analyze the data and identify underlying factors that influence veterinarians' antimicrobial prescription behavior. We found that under "Capability," factors such as knowledge of antimicrobial resistance (AMR); ability to handle complex disease conditions; ability to identify the appropriate antimicrobial type, routes of administration, and potential side effects influence prescription behavior by veterinarians. Under "Opportunity," veterinarians' prescription behavior was influenced by lack of laboratory testing facilities, poor farm biosecurity, farm management and location, farming conditions, impacts of climate change, the clinical history of animals and social influence from different actors

study did not give consent to make their interview transcripts available in the public domain. Additionally, the transcript may contain some sensitive information such as the brand name of the antimicrobials, as well as participants could be identifiable because of their professional affiliation, which may put them at risk. However, data request can be made to the corresponding author at mkalam@emory.edu or to the ethical committee directly at drecvasu@gmail.com. An anonymized dataset can be available with an appropriate purpose.

**Funding:** This work was funded by Bangladesh Bureau of Educational Information and Statistics (BANBEIS), ID number SD2019967. Mohammad Mahmudul Hassan was supported by BANBEIS. The funders had no role in study design, data collection and analysis, decision to publish, or preparation of the manuscript.

**Competing interests:** The authors have declared that no competing interests exist.

including senior figures, peers, farmers, and other informal stakeholders. Under "Motivation," national laws and guidelines serve as catalysts in reducing antimicrobial prescriptions. However, perceived consequences such as fear of treatment failure, losing clients, farmers' reliance on informal service providers, and economic losses demotivate veterinarians from reducing the prescription of antimicrobials. Additionally, veterinarians feel that reducing the burden of AMR is a shared responsibility since many informal stakeholders are involved in the administration and purchase of these medicines. Based on our results, this study recommends incorporating the factors we identified into existing or novel AMS interventions. The behavior change wheel can be used as the guiding principle while designing AMS interventions to increase capability, opportunity and motivation to reduce antimicrobial over-prescription.

## 1. Introduction

Antimicrobial resistance (AMR) is considered as an invisible pandemic because of its immense but slow-moving effects on mortality, morbidity, and health systems [1,2]. Rooted in the excessive use of antimicrobials in both human and animal populations, AMR can propagate through ecosystems, rendering widely available antibiotic therapies used to treat microbial infections in humans and animals ineffective [3–5]. Evidence shows that the use of antimicrobial agents is substantially higher in food-producing animals than in humans, which presents a major threat for human, animal and environmental well-being [6,7]. Due to multifaceted and diversified structural, cultural, socioeconomic, and systemic challenges, lower- and-middle-income countries (LMICs) have a higher burden of AMR [8]. In Bangladesh, as in the case of other LMICs, these factors may increase the use of antimicrobials in commercial animal farms and are thought to be responsible for the development and spread of AMR in pathogens [7,9,10].

While the optimal use of antibiotics is important for animal welfare, thereby for protecting farmers' livelihoods [11,12], their overuse potentially selects for AMR bacteria that can propagate up the food chain to humans through the consumption of animal-source foods or through environmental contamination [12,13]. A systematic review documented the prevalence of different types of antimicrobial-resistant pathogens such as *Salmonella enterica*, *Escherichia coli*, *Pseudomonas aeruginosa*, *Staphylococcus spp.*, and *Bacillus spp.* in poultry and dairy farms in Bangladesh [10]. Therefore, Bangladesh is currently faced with an emerging AMR problem through animal-source protein such as poultry, eggs, meat, milk, and milk products [7,10]. As in the case of human antibiotics in Bangladesh [14,15], one of the challenges in combatting AMR in food-producing animals lies in the largely unregulated access to, delivery, and use of, antimicrobials. Over-the-counter (OTC) sale of antimicrobials from drug shops (for both human and animal use) has been identified as one of the main causes of misuse or overuse of antimicrobials in Bangladesh [16–18]. This occurs even though the Bangladesh Veterinary Practitioners Ordinance 1982 specifies that only registered veterinarians are permitted to prescribe medication or perform surgery on animals [19]. Similarly, according to the Drug Act of 1940 [20], only registered pharmacists are authorized to provide antibiotics with a valid prescription.

Registered veterinarians in Bangladesh can play a pivotal role in promoting prudent, safe and optimal antimicrobial use (AMU) in food-producing animals [21] because of their direct relationships with farmers, pharmaceutical companies, and other stakeholders [22,23].

Previous research has identified factors influencing veterinarians' antimicrobial prescription decisions, including resource availability, farm setting, veterinarians' knowledge, awareness, and attitudes towards antimicrobials, and influential relationships (e.g., with pharmaceutical companies) [21,24–26]. However, while these studies have made strides in quantifying various dimensions of antimicrobial prescriptions, they are unable to elucidate or offer contextual and underlying insights regarding the factors influencing antimicrobial prescription behavior by the registered veterinarians. A recent review analyzing 103 research articles shows that knowledge and awareness of antimicrobials, attitudes towards antimicrobials, influential relationships, resources, and factors affecting antimicrobial use are the main behavioral factors that globally influence antimicrobials prescription decisions among registered veterinarians [27].

Antimicrobial stewardship (AMS) programs—efforts aimed at optimizing antimicrobial use in human and veterinary medicines—have been found to be effective strategies in combating AMR [28]. In order to achieve sustainable and tailored results, it is critical to include veterinarians in current AMS programs that are theoretically driven, achievable, and locally appropriate [32]. Because of the ability to address structural, psychological, social, and cultural drivers of antimicrobial overuse, behavioral and implementation scientists have proposed that employing behavioral frameworks and theories could bring sustainable changes in veterinarians' prescription behaviors [29–32]. Additionally, behavioral interventions are easy to adapt, replicate and scale-up across lower-resource settings. However, before designing or adapting a behavioral intervention, understanding, or exploring the key behavioral drivers of antimicrobial prescription by the veterinarians is the first step. In Bangladesh, factors associated with antimicrobial prescribing behavior of veterinarians is poorly understood [33]. To fill this gap, the aim of this study was to identify the factors associated with antimicrobial prescribing behavior among registered veterinarians in Bangladesh through the use of the Capability, Opportunity and Motivation for Behavior (COM-B) framework. Before explaining the study's methodology, a conceptual framework is described next to provide a better understanding of the theoretical framework of this study.

## 2. The conceptual framework

We employed the Capability, Opportunity, Motivation and Behavior (COM-B) model to identify factors that influence prescription behaviors. Developed based on an extensive review of existing behavior change theories and consultation with the behavior change experts, the COM-B model offers an outcomes-based approach to the study of human behavior. It perceives human behaviors as outcomes that are influenced by factors at different levels and helps devise comprehensive guidelines for studying human behavior [34]. The term '*Capability*' refers to an individual's capacity to engage in behavior modifications, while '*Opportunity*' pertains to the environmental factors influencing individual behaviors. '*Motivation*' denotes the brain processes that energize and direct behavior and includes processes, emotional responding and analytical decision making for change [34]. Expanding on these three domains, the COM-B model further divides into six sub-domains (Fig 1). To uncover underlying and nuances of the factors, efforts have been made to assemble the Theoretical Domains Framework (TDF) and the domains and sub-domains of COM-B model [35]. Comprising 14 domains, the TDF offers a comprehensive assemble of the overlapping constructs within behavioral theories and provides explanation of each domain [35,36].

The COM-B model has been used widely in designing and implementing various health interventions in clinical and community-based settings by implementation scientists globally [34]. Numerous studies have explored factors affecting human antibiotic use behavior across lower, middle- and upper-income countries using this model [37–40]. However, until recently,

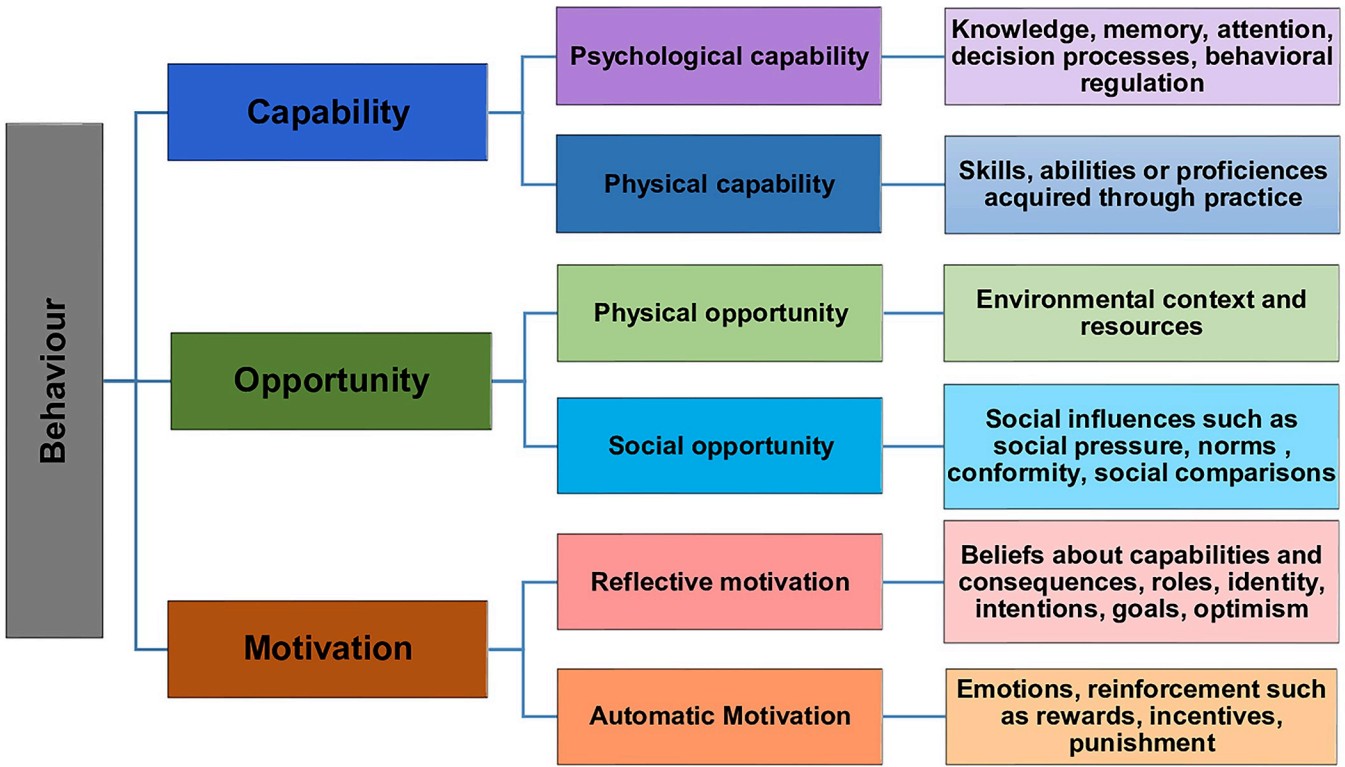

**Fig 1. Conceptual framework: Integration of Theoretical Domains Framework (TDF) into the corresponding constructs of the COM-B model.**

application of this model in optimizing veterinary antimicrobial prescription and use behavior has been very limited. A recent Irish study [37] and a systematic review [41] noted that the use of COM-B model could be better suited to bring about sustainable changes in veterinary antimicrobial use.

## 3. Materials and methods

### Study design

We employed an explorative qualitative design in this study. A qualitative approach was considered suitable for uncovering the nuances in the domains of the COM-B model in studying the socio-cultural aspects of prescribing antimicrobials. Where applicable, the study followed the consolidated criteria for reporting qualitative research (COREQ) guidelines of reporting qualitative studies (S1 File) [42].

### 3.2 Sampling, and recruitment procedure

We used a purposive sampling strategy to include veterinarians in the study for the interview. We invited veterinarians who met the following inclusion criteria, i) practicing veterinarians, ii) those involved in treating poultry, livestock, and other food-producing animals. Before inviting them to the study, the authors (SS, MAK, and MMH) collected a list consisting of veterinarians from the Bangladesh Veterinary Council as well as intern veterinarians. From this list, we created sub-lists based on profiles and different characteristics such as gender, years of practice (experience), practice locations (geographic distributions—urban or rural), nature of affiliations (government vs non-government), and educational qualification (bachelor, masters

or higher). We recruited participants from all these sub-groups to capture nuances and diversity in the data, aimed at ensuring the validity of our findings.

### 3.3 Interview guide

A semi structured interview guide was developed based on the literature. The interview guide primarily included questions related to general views and practices regarding animal health management, approaches to providing services to animals, knowledge and awareness about AMR, considerations in prescribing antimicrobials, barriers, and facilitators in reducing the burden of AMR, and professional role in addressing the AMR problem. The initial version of the interview guide was then reviewed by a group of experts, including veterinary researchers, practitioners, public health researchers, and university academics. Based on their comments and feedback, we revised the guideline. Subsequently, we translated the guidelines in Bangla (the local language). The Bangla version was back translated into English to compare the changes and deviations. We had to make some modifications to the original version based on this process. Once the research team reached a consensus on both versions, we piloted the interview guide with 4 participants (these 4 interviews were not included in this analysis)—checking for data suitability and validity, and comprehension of the data collectors. The pilot interviews also helped determine probing and additional questions. A slight modification was made to the Bangla version based on the pilot exercise. These modifications were incorporated into the final version.

### 3.4 Ethical approval and data collection

Ethical approval was granted by the Research Ethics Committee of Chittagong Veterinary and Animal Sciences University (CVASU), Chittagong, Bangladesh [approval number CVASU/Dir (R&E)/EC/2020/169(3), dated 29 December 2020]. We conducted this study in accordance with human subject research guidelines set forth in the Declaration of Helsinki 1964 and its later amendments. In order to conduct the interviews, we recruited participants between 4 May to 29 June 2021. A written informed consent was obtained from each participant before starting the interview, and the written consent was recorded in a form where both participants and interviewers put their signatures. After putting the signature, the interviewers kept one copy of written consent for the study record and gave a copy to the participants for their own record. Witness signature was not required for this study because the participants were highly educated, adult and able to make informed decision to take participate in the study. The participants had the freedom to withdraw themselves from the study at any stage of the interview. The interviews were recorded in a digital audio recorder after obtaining the written consent. A team of trained and experienced research assistants (MSR, SA, MNU, and FAJ) with an academic background in Anthropology conducted the interviews. The interviews were conducted in person, by following the COVID-19 safety protocols established by the Directorate General of Health Services in Bangladesh. All the interviews were conducted in participants' workplace which ensured the privacy of the discussion and biased free.

### 3.5 Data analysis

We carried out thematic analysis of the interview transcripts using both deductive and inductive codes with MAXQDA Standard (2020, VERBI Software, Berlin, Germany). Before importing into the software, the transcripts were first anonymized, and a unique code number was assigned to each interview. Initially, all the recorded interviews were transcribed into Bangla (the local language) and then cross-checked against the audio recordings for

accuracy and completeness. Subsequently, the interview transcripts were translated into English by the team. The authors (SS, MAK, SA and FAJ) reviewed and ensured the quality of the translations. Data familiarization was achieved by thoroughly listening to the audio recordings and repeatedly reading the transcripts. During this stage, researchers developed deductive codes based on the domains, sub-domains, and constructs of TDF and COM-B model. To create the inductive codes, three authors (SS, MAK and SA) read two transcripts and, through consensus, generated a preliminary set of open codes derived from the texts. These codes (both deductive and inductive) were then applied to guide the analysis of the remaining transcripts, with additional inductive codes introduced whenever new insights emerged. Based on the nature of the codes, we subsequently categorized and clustered the codes into different themes and sub-themes. The identified overarching themes and sub-themes were first grouped per domain of TDF and then these TDF domains were further categorized and grounded in the COM-B model. In the final stage, we reviewed all transcripts once again and employed axial coding techniques to ensure that the identified themes and sub-themes were logically and coherently aligned from an analytical perspective.

## 4. Results

### 4.1 Characteristics of participants

We enrolled 32 participants in the study. Characteristics of the study participants are provided in Table 1. The majority of the participants were male (n = 29) and most participant completed a Masters' degree or higher. The study included participants with different levels of experience, and most were affiliated with government veterinary practices. The majority of the participants had received training on AMR (n = 26).

**Table 1. Participants' characteristics.**

| Characteristics of the participants | | Number |
|---|---|---|
| Gender | Female | 3 |
| | Male | 29 |
| Age | 26–35 | 8 |
| | 36–45 | 12 |
| | 46–55 | 10 |
| | 55 or more | 2 |
| Educational attainment | DVM* | 8 |
| | Masters' and postgraduate | 24 |
| Experience (in years) | 1–5 | 6 |
| | 6–10 | 8 |
| | 11–15 | 10 |
| | 16–20 | 5 |
| | 21 or more | 3 |
| Professional affiliation | Government | 22 |
| | Private corporation | 5 |
| | Freelance** | 5 |
| Training on AMR | Yes | 26 |
| | No | 6 |

*Doctor of Veterinary Medicine (bachelor level education).

**Independent practitioners who are not formally affiliated with any organizations.

**Table 2. Summary of the results in line with COM-B constructs and TDF domains: Retrieved from the inductive analysis.**

| COM-B Constructs | | TDF domains | Findings from the interviews |
|---|---|---|---|
| Capability | Psychological | Knowledge, memory, attention, decision process, behavioral regulation | Knowledge on AMR is fairly good among the participants. |
| | | | Lack of knowledge about the reserve group of antimicrobials among the young and recently graduated veterinarians. |
| | Physical | Skills, abilities or proficiencies acquired through practice. | Based on the field experience veterinarians with 15 or more years can handle complex disease condition of the animals while recently graduated or young veterinarians rely on the seniors. |
| | | | Ability to identify the type of antimicrobials, route of administration and potential side effects influence veterinarians to suggest different type of antimicrobials. |
| Opportunity | Physical | Environmental Context and resources | Lack of technical facilities such as laboratory testing. |
| | | | Facilities related to farm management, biosecurity, farm location, farm population and farm size. |
| | | | Context of avoiding secondary infection. |
| | | | Climate change and weather. |
| | | | Clinical history of the animals. |
| | Social | Social influences such as social pressure, norms, conformity, social comparisons | Influence by the senior veterinarians. |
| | | | Influence by the peers. |
| | | | Influence by the farmers. |
| | | | Influence by other social stakeholders such as agents from pharmaceutical companies, quacks (informal service providers) and feed and drug sellers. |
| Motivation | Reflective motivation | Beliefs about capabilities and consequences, roles, identity, intentions, goals, optimism | Veterinarians are well accepted by different social groups including farmers. |
| | | | Farmers may rely on informal service providers or drug and feed sellers if the vets reduce the prescription of antimicrobials. |
| | | | Vets fear of losing clients if they reduce the prescription of antimicrobials. |
| | | | Vets are fear about the failure of treatment which may ultimately impact on their reputation. |
| | | | Farmers may face economic loss as a result of increased disease or death of the animals. |
| | Automatic motivation | Emotions, reinforcement such as rewards, incentives, punishment | National law and guidelines empower veterinarians to promote reduced use of antimicrobials. |
| | | | Vets closely work with different stakeholders which provides motivation to aware on AMR. |

## 4.2 Thematic results

This paper focused on the factors that influence the veterinarians' antimicrobial prescription behavior. The results are organized based on the COM-B model and the domain of TDF. The factors influencing veterinarians' antimicrobial prescribing behavior are shown in Table 2. A number of domains were identified including knowledge; skills; beliefs about consequences; social/professional role and identity; social influences; environmental context and resources.

**4.2.1 Veterinarians' CAPABILITY to promote reduced prescription of antimicrobials.** *Knowledge on AMR.* All participants showed an awareness about the relationship between the misuse or overuse of antimicrobials and the development and spread of AMR. We also found that they were aware of national guidelines on judicial use and different groups of antimicrobials. Analysis revealed that their knowledge of AMR and its drivers were guided by their academic background, practicing experience in the sector and participation in different training and workshops. When gauging participants' knowledge about how AMR evolves, several pathways were identified. A commonly identified pathway pertained to ineffective drug treatments; specifically, the concept that when drugs lose the ability to stop bacterial growth,

bacteria may become resistant, leading to serious infection. Additionally, some of the participants said that antibiotic withdrawal periods should be respected, otherwise antibiotic residues could be consumed and exert selection pressure among human [gut] bacteria. One participant mentioned that,

*"Suppose a farmer will sell chicken on the 30th day. On the 27th day, their chicken developed a cold problem. In that case, when they want to use antibiotics, we cannot allow it. Because they will sell his chickens on the 30th day. What will happen if I give them antibiotics at that time? Antibiotics will remain in the flesh. It will remain [in the] legs, chest, skin, and liver. When people cook and eat that chicken, antibiotic resistance will be created in the human body."* (Participant—3)

Furthermore, they discussed that selection for AMR can lead to the complete failure of medicines. After a certain time, antimicrobials will not work if the antimicrobial/drug selected for treatment is not appropriate to kill the bacteria.

*Knowledge about the reserve group of antimicrobials.* Reserve group antimicrobials are considered as the last resort that can be used for severe infections caused by AMR. In this study, our findings suggest that experienced (more than 15 year of practicing experience) and participants who received AMR training had fair knowledge about those reserve group of antimicrobials. In contrast, those who are relatively new to practice were not be able to articulate about these medicines. Among them, one participant said,

*"I heard that there are some reserve groups of medicines. However, I am not aware why these medicines are reserved–but I guess these are not effective to use or treat or may be for future use".* (Participant– 7)

*Ability to differentiate groups of antimicrobials and their use.* In general, experienced veterinarians could easily identify different types of antimicrobials and respective uses while juniors or recently graduated veterinarians expressed a lack of perceived competence in determining the type of antimicrobials. Veterinarians suggested antimicrobials based on their theoretical knowledge and practical experience. Thus, more experienced veterinarians found it easier to suggest appropriate antimicrobials. One participant with 12 years of field experience mentioned,

*For mixed bacterial cases like- Salmonella, E. coli, Mycoplasma, Fowl cholera infections vets use broad spectrum including ciprofloxacin. But theoretically only bacterial disease should influence prescribing behaviors of antimicrobials, most veterinary surgeons, like me, prescribe antimicrobials as a supportive treatment and following the signs and symptoms of viral diseases. Viral disease played a role in prescribing behaviors.* (Participant– 14)

Another participant said,

*". . . . .and in the case of viral diseases such as FMD [foot and mouth disease], ND [Newcastle disease]and AIV [Avian Influenza Virus], there is usually no reason to use antibiotics. But for some reason, antibiotics are used as a supportive treatment. For example, in case of viral diseases, antibiotics are used if there are any signs or symptoms and in case of bacterial diseases, antibiotics are used well. It is seen that one to two antibiotics are given in each prescription."* (Participant– 1)

FMD is a viral disease, but veterinarians use antimicrobials to lower mortality risk. Antimicrobials cannot directly target FMD but they suggest these medicines to mitigate secondary (bacterial) infection, weakness, and mortality rate.

*Factors influencing choice of antimicrobials.* The findings suggest that several factors influence veterinarians' choice of antimicrobial, including ease of administration and potential side effects. Generally, participants preferred food based or injectable antimicrobials. One participant said,

> *"You know there a variety of options. But I prefer food based and injectable ones. For the food-based antibiotics are easy to administer for the farmers because they do not require to build expertise to administer. All they need is to feed the medicines with foods".* (Participant– 19)

Some veterinarians preferred injectable antimicrobials for large animals like bovines or goats. One participant said,

> *"I prefer injectable because it is easier for me to use. If I do not have the regular visit to the farm, the farmer can hire someone from the nearby to inject it."* (Participant 4)

**4.2.2 Veterinarians' OPPORTUNITY to reduce prescription of antimicrobials. Technical facilities.** Almost all of the participants mentioned that not having culture and sensitivity (CS) test facilities is a major barrier for promoting safe use of antimicrobials. This is particularly a major problem in rural and peri-urban areas where the veterinarians are practicing with no proper testing facilities. One of the participants said,

> *"If the drug is used after CS test, it will work 100%. Less than 100% probability of antimicrobial resistance occurred. If you go to the vets and they will give a CS test to see which organism is sensitive to an antimicrobial and that antimicrobial has been matched. Unfortunately, in rural areas, there is no such facility".* (Participant– 5)

Participants believed that having those facilities would enable them to prescribe antimicrobials appropriately. Specifically, nineteen (19) participants mentioned that in rural and remote areas, facilities such as lab testing machines, chemicals, or development of veterinary infrastructure and equipment are required to promote for safe use of antimicrobials.

It was also noted by the participants that, sometimes owners from remote areas are unwilling to visit veterinary hospital/veterinarians at the Upazila level [the lowest administrative unit of the country] because of hassles in transportation. It was also discussed by the participants that diagnostic facilities are available in the city or urban areas, but less in the peri-urban and rural areas. However, the majority of farms are located in rural areas, which are usually distant from facilities. As a result, farmers feel reluctant to bring their large animals (for example ruminants or goats) due to the transportation hassle and cost. One participant said,

> *". . . . .this is particularly a huge barrier for the farmers those who have cows or goats due to the long distance and cost. Because farmers have to hire a transportation like motorized van which costs a lot of money."* (Participant– 12)

In all cases such as the absence of technical facilities, structural and financial challenges and farmers' reluctance to bring the animals to the vets which limits the opportunity to prescribe antimicrobials appropriately.

**Facilities related to farm management.** Improved or proper farm management can reduce the disease burden and is considered an essential driver to prevent antimicrobial use and AMR. Proper farm management initiatives include ensuring a proper dietary system, nutritional supplements, and immunization of the animals. One of the participants mentioned,

*"Food has a vital relation with immunity boost up and nutrition. Most of the time poor farm management hampered the feed quality of the animals. Cattle and poultry are infected due to lack of proper nutrition and dietary system".* (Participant– 21)

Maintaining proper biosecurity of the farms is another vital capability that helps reduce the overuse of antimicrobials. A farm with proper biosecurity can prevent viral or bacterial infections. According to the veterinarians, farms without proper biosecurity are usually affected by bacterial and viral infections. In an improperly bio-secured farm, humans, domestic animals, and wild animals can enter easily and become carriers of several diseases and bacteria. According to the veterinarians, in most of the farms, there are no control sheds or hygiene facilities before entering the farms. People are unaware of maintaining proper disinfection and hygiene procedures at the entrance of the farms. As a result, animals are getting infected with different viral and bacterial pathogens. One of the participants said,

*"Farm biosecurity and hygiene need to be maintained properly. Sometimes, the farms are affected by different viral and bacterial infections that carried by humans and animals. This is also a major reason for over prescription of antimicrobials in the farms".* (Participant– 15)

Flock size is one of the major factors affecting antimicrobials prescribing by the veterinarians. Many participants claimed that in their practicing areas, farmers do not follow any recommendations in determining the size of the chicken flock considering the land size. Rather, they believe that livestock owners focus on money and want to raise more chickens per land area to minimize the cost while maximizing the profit. However, large herds or flocks kept in smaller spaces have a greater chance of infection and veterinarians are required to prescribe antimicrobials considering the farm population. One participant mentioned,

*"In densely populated farms vets prescribe AM [antimicrobials] to reduce the infections and mortality rate. But veterinarians always think about the overdose of AM [antimicrobials] while prescribing the doses".* (Participant– 12)

**Animals' clinical history context.** The previous history of using antimicrobials is considered a vital factor for prescribing. Veterinarians usually ensure that they collect the medical history of the animals. In the case that they realize a certain antimicrobial is successful in controlling infections, they usually prescribe the same antibiotics again. A participant said,

*"Definitely we collect animal history from the owner because they are my regular [permanent] client. If we see that a particular antibiotic worked in the past, I will definitely choose that one because I would not take the risk to try a new one".* (Participant– 18)

However, when an animal's previous records are unavailable, veterinarians prefer to use higher doses of the most recently administered of antimicrobials to that specific animal because they are unsure of what type of medicines were previously prescribed. In many cases, owners cannot properly report the previous use of antimicrobials, and veterinarians find no

choice but prescribe higher doses because lower doses are perceived ineffective due to possibly already prevalent AMR from excessive and inappropriate use of antimicrobials. A participant mentioned,

> *"Owners cannot tell the history of the animals properly. Sometimes they hide the information about using antimicrobials because they already started the medicine on their own or suggested by the quack [informal service providers who do not have formal education and license to practice]. In most cases, I do not have any choice but rather use double doses of antimicrobials".* (Participant– 9)

**Context of avoiding secondary infections.** A cluster of explanations implied that antimicrobials were prescribed as a supportive treatment, particularly by following the signs and symptoms of the diseases. Despite knowing the fact that antimicrobials are not effective for a viral infection like Newcastle disease (ND), they prescribe antibiotics due to a perceived risk of secondary infection that is feared to increase mortality rates in the animals. Some diseases like ND and Gumboro which are viral diseases but may cause secondary bacterial infections and require antibiotic doses. One participant said,

> *"FMD [Foot-and-mouth disease] is a viral disease, but for FMD case, we give it antimicrobials to stop secondary infection. So that the cow does not die. You cannot give any antibiotics against this virus. But we give antibiotics so that they do not die due to secondary infection".* (Participant– 14)

During the rainy season, domestic animals (like, cattle, sheep, and goats) may get infected by diseases like anthrax. As a preventive measure against some seasonal diseases and changes, veterinarians' resort to prescribing antimicrobials. They also considered prescribing during brooding period when they prescribe antimicrobials as a precautionary measure. For example, during the brooding period on the very first day, veterinarians prescribe antimicrobials for three days to prevent diseases and bacterial infections. The doses are maintained according to the immune system of the birds. One participant mentioned,

> *"During the first three days, we prescribe antimicrobials, as so that no disease can attack at the beginning. But it is against clinical judgment. Because there is no disease, and the birds are clinically healthy and fit. Thinking only of the future, we prescribe for three days. In that case, we customize the doses according to the condition of the birds. If the birds are normal, we prescribe normal doses. But if the birds have any problem, we increase the doses."* (Participant– 19)

**Climate change/weather—an emerging environmental context.** A few veterinarians (n = 7) mentioned that climate change or weather condition is another factor that influences their antimicrobial prescription. High humidity is cited as the reason for spreading infections among poultry/birds in Bangladesh. This climate change issue increases the mortality rate in birds. One participant mentioned that,

> *"High humidity and humid conditions are the reasons for spreading infections among birds, which increases the mortality rate. If we maintain the airflow properly, we can control the humidity. But it is a very expensive and most of the farmers do not have the affordability to purchase these machineries".* (Participant– 21)

Fluctuations in temperature and seasonal changes are considered major problems for maintaining a proper farm environment. The veterinarians indicated that farmers are not aware of the way seasonal changes affect humidity and are not equipped to control the temperature. So, they (owners) were not able to prevent the spread of diseases and infection rates.

*Social Influence.* **Influence of senior vets:** A cluster of responses from newly graduated vets indicated that participants were often influenced by their peers, senior colleagues, or authority figures (in the case of large farming companies) to prescribe antimicrobials. Analysis indicates junior veterinarians are more likely to be guided by senior vets who are experienced in the field. One vet stated,

*Recently I visited a farm along with my senior. The chickens suffered by viral diseases. I understood antimicrobials are not applicable for this case. However, my senior approached me to suggest antimicrobials. . . . . . . . .[later]. . . . . .. Sometimes I call over the phone to my senior [the veterinary surgeon] to discuss and find a right antimicrobial. After describing the condition, my senior suggest an antimicrobial and I prescribe it to the farmers".* (Participant– 9)

**Influence of peers.**   A number of participants (n = 21) stated that they discuss disease conditions and exchange ideas among themselves in different professional forums, workshops and in personal communication. Additionally, group discussions are held when several vets share a locality. In the case of successful use of antimicrobials against a given disease condition, vets suggest the same antimicrobials to others. One participant stated,

*"You know most often we [the vets] come together and discuss. At that time, we discuss about different aspects of our practice and share our experiences. If someone says I used a particular antimicrobial for a particular disease and found successful, the other vets would definitely rely on that treatment".* (Participant– 31)

**Influence of farmers.**   It was also reported that farmers prefer not to maintain long courses of treatment for their animals, instead desiring medicines that work rapidly. One widespread perception among the veterinarians is that vets who do not suggest antibiotics are not good vets. Most often, owners directly administer antimicrobials on their own or have them administered by quacks without maintaining the doses and treatment protocol. As a result, vets mentioned, if they do not suggest antimicrobials then the farmers would use antimicrobials by themselves or go to the quacks to get suggestions. In this case, things will be a difficult to control. Therefore, sometimes vets have to suggest antimicrobials so farmers can use a right antibiotic with a right dose. One participant said,

*"Farmers come to us to get antimicrobials because they want to see the quick results. If we do not give antibiotics, they would go to the medicine shop directly or to the quacks who do not have the medical knowledge or skills to use antibiotics. Sometimes, we have to suggest antibiotics, to prevent that misuse. If we suggest, at least they can get the right antibiotics with a proper duration, and they follow the course".* (Participant– 16)

Several vets (n = 8) also mentioned that sometimes they suggest antimicrobials fearing loss of clients. They try to retain clients by meeting farmers' requests seek their services. One participant said,

*"Farmers come and ask to give more medicines. They do not try to understand that medicines have side effect and should be given as needed. Nevertheless, we have to prescribe for their [farmers] satisfaction".* (Participant– 12)

**Influence of other social actors.**   A number of participants reported other social actors such as the authority (veterinarians who are working in the private companies), pharmaceutical companies, and local sellers of animal feeds and medicine. Vets working for large companies said,

*"The use of antimicrobials has some economic aspect of the corporations because they produce poultry in large volume."* (Participant– 9)

Some explanations indicated that *"fear of losing job"* is a common reason that led to suggesting use of antimicrobials.

Several participants mentioned that pharmaceutical sales representatives provide antimicrobials to the vets and ask them to suggest to the farmers. Vets are also enticed to use new groups or brands of antimicrobials. One participant said,

*"Sometimes we use a new drug experimentally and when we get a good result, we use these regularly".* (Participant– 7)

**4.2.3 Veterinarians' MOTIVATION to promote reduced antimicrobial use/act as AMR stewards.**   *Role in AMR stewardship.* In this study, all participants stated that according to the national law, policies and guidelines, veterinarians are responsible for prescribing antimicrobials. At the same time, they are also accountable for proper and appropriate administration of these medicines. However, as they are operating in a complex socio-cultural landscape with poor veterinary infrastructure (i.e., lack of testing capacity for animal-borne pathogens) and the influence of many other actors (i.e., drug and food vendors, informal practitioners, etc.) for acquiring and administering the antimicrobials, stewardship is a shared responsibility for all stakeholders. Participants in this study stated that veterinarians need more knowledge and skills in order to find appropriate antibiotics for a given situation. Some also felt the need for revising treatment protocol and guidelines based on changing disease patterns and discovery (and availability) of new antimicrobials. One participant stated,

*"Veterinarians can be acted as the catalyst, change maker and lead in reducing the use of antimicrobials. But they need more updated knowledge. Special training or workshop can be conducted with the veterinarians".* (Participant– 15)

Some participants also suggested that vets could be involved with awareness raising among the farmers because they directly engage with them. Because of their respected position in animal welfare, and in the veterinary landscape, vets are esteemed among farmers and other actors. Their status and reliability can be harnessed in awareness-raising. One participant mentioned that,

*"Vets can definitely be engaged in raising awareness. Basically, many of us aware the farmers. We need to make it clear to the farmers that, we are going to be in a dire situation or that we are facing long term losses if they use antimicrobials unnecessary. A massive campaign should be carried out especially with government and private initiatives".* (Participant– 11)

Some participants urged policymakers to update the national treatment protocol and guidelines based on the contemporary disease conditions and availability of treatment facilities. In this sense, the veterinary council needs to share proper guidelines and treatment protocols for prescription. One participant stated,

*"The guidelines need to be updated based on the current condition. The veterinary council should play an active role to advance contemporary knowledge and updates according to the new changing conditions".* (Participant– 21)

Additionally, participants also recommended that vets be involved in promoting awareness about the scientific management of farms, including vaccination, alternative medicines like probiotics, and minimizing the influence of quacks in the community in order to reduce use of antimicrobials.

*Professional identity and social position.* By national law, veterinarians are responsible for prescribing antimicrobials and mandated with operating based on national guidelines and treatment strategy in order to provide veterinary health services in Bangladesh. Because of their professional expertise (as the legal and formal responsibility holder to treat animals), they are highly accepted by farmers and other relevant stakeholders in the community. One participant said,

*"Look, we are not only responsible for prescribing antimicrobials to the animals, we are also responsible for human health. Unjudicial use of antimicrobials is a threat to human, animal and environmental health. As a vet, we are responsible to limit the use of antibiotics."* (Participant– 18)

This professional demeanor is one of the motivating factors that promotes responsible prescribing of antimicrobials. The sense of responsibility derived from their professional identity puts them in an influential position among farmers and other relevant stakeholders. One participant said,

*"Farmers come to us to seek services. We work with the farmers closely and they [farmers] follow our suggestion and recommendation".* (Participant– 23)

Antimicrobial stewardship efforts should therefore leverage vets' professional expertise and social position in animal welfare.

*Beliefs about consequences of reduced prescription of antimicrobials.* Participants noted both positive and negative consequences of reduced use of antimicrobials. Negative consequences included increased disease and economic loss. One participant said,

*"In rural areas, the biosecurity condition is not up to the mark. Farm's hygiene condition is very poor. Due to these factors, many diseases affect the animals, and we have to prescribe antimicrobials. So, if we reduce the use of antimicrobials then there is a chance to increase disease risk".* (Participant– 9)

Due to an increased disease risk, there is a high chance of economic loss on the farm. However, participants also mentioned positive consequences of reduced use of antimicrobials. These included protection of animal, human and environmental health, reduced spread of AMR pathogens and lower prevalence of resistant bacteria. One participant mentioned that,

*"Overuse of antimicrobials is the major reason for spreading AMR infection to human. It is also problematic for the environment. When we can reduce the use of these medicines, we can protect humans, animals, and the environment. That's why AMR is a One Health concern."* (Participant– 21)

Some veterinarians stated that there is also a fear of treatment failure resulting from reduction in prescription of antimicrobials and due to the unsuccessful treatment veterinarians may lose their reputation. This perception was recorded among the young veterinarians. One participant said,

*"For example- we suggest one treatment with some basic medicines, but animals did not respond to the treatment and the condition may become severe. If the condition becomes severe, the owners may face economic loss and we will lose our reputation".* (Participant– 8)

A few participants also stated that if the vets reduce the prescription of the antimicrobials, then the farmers may rely on informal quacks or feed and medicine shop owners to get these medicines. A participant said,

*"If the vets start reducing prescribing antimicrobials, then the farmers would go to the quacks. They can also get these medicines from the medicine shop owners, and even from the feed sellers".* (Participant– 28)

*Optimism to promote reduced prescription of antimicrobials.* Several participants were not fully optimistic about promoting reduced prescribing of antimicrobials due to existing veterinary infrastructure, the complexity of field practice, and lack of monitoring of inappropriate and informal sale of these medicines. One participant said,

*"We are working blindly. Means when we write a prescription, we cannot perform any tests and that limits our proper prescription opportunity".* (Participant– 13)

Relating to the complexity of the working environment, another participant said,

*"I work in a rural area where people mostly rely on traditional quacks. Quacks on the other hand do not follow any scientific process. When they fail to treat animals, owners come to me. In the meantime, the damaged has been made and I cannot undo it and I have to prescribe antimicrobials to prevent the death of the animal or economic loss of the owners".* (Participant– 16)

In addition to that, people can purchase antimicrobials without any prescription over the counter. Veterinary drug and feed sellers also give antimicrobials. One participant said,

*"Even if we reduce prescribe antimicrobials, it would not work because there are many people are operating here who give antimicrobials without following any process. Without changing their behaviors, it is difficult to bring change".* (Participant– 31)

Participants also mentioned that there is no active or proper monitoring on the sale and exchange of these medicines by these informal stakeholders who are legally not authorized to suggest antimicrobials.

## 5. Discussion

To the best of our knowledge, this is the first study that has applied the COM-B model to explore factors behind veterinarians' antimicrobial prescribing behaviors in a LMIC. In line with the COM-B model, this qualitative study identified different factors that determine veterinarians' prescription behaviors. Findings indicated that factors fell into several TDF domains, namely knowledge, skills, social/professional role, and identity; beliefs about capabilities; beliefs about consequences, motivation and goals, decision processes; environmental context and resources; social influences; and action planning. Inductive inferences under each domain of TDF and COM-B model revealed the intricacies of prescribing antimicrobials to food-producing animals in Bangladesh.

In this study, AMR knowledge was identified as an influencer of veterinarians' capability to appropriately suggest antimicrobials. All the participants in this study completed tertiary-level education while some pursued advanced degrees in the field. Additionally, participation in different scientific workshops, conferences, and trainings enhanced their knowledge and skills. These findings aligned with earlier studies in Bangladesh [18,21] and those in similar LMICs [43–46]. However, the experiential narratives indicate that the selection of antimicrobials was not always clear among the study veterinarians. Some veterinarians, particularly those who were recent graduates and were relatively new to practice, lacked adequate knowledge on the reserve group (last resort) of antibiotics. This finding supports an earlier cross-sectional study in Bangladesh [21]. These reserve antimicrobials are considered by some vets to be 'unnecessary' or inappropriate [47]. Participants in this study also felt the need for continued educational training to acquire contemporary and up-to-date strategies to address changing animal disease patterns. This is particularly critical to reduce antimicrobial prescriptions because when an unusual disease is seen in animals, vets tend to prescribe antimicrobials without performing any laboratory tests [48].

Broadly, vets' opportunity (the external factors) to promote reduced antimicrobial use was impacted by physical opportunities and social opportunities. Under the physical opportunities domain, our inductive findings showed that lack of technical facilities to perform antimicrobial sensitivity testing, poor farm management practices (inadequate biosecurity, lack of farm population optimization, and non-ideal location), context of animals' clinical history, context of avoiding secondary infection, and climate change impacted vets' prescription behaviors. As per TDF, these inductive findings are grouped under environmental context and resources domains. Our findings on factors related to physical opportunities are aligned with existing international literature. Prior studies have shown that a lack of rapid and effective diagnostic techniques may support the development of resistant microorganisms [49]. A higher cost of culture and sensitivity testing led to the inappropriate use of antimicrobials by livestock professionals in Bangladesh [17]. Additionally, under the physical opportunity domain, other factors that influenced vets' prescription behaviors was the focus on avoiding secondary infections in possible but clinically unconfirmed cases of viral diseases. Prior studies in Europe, Australia and different LMICs including Bangladesh, India, and Thailand have also reported similar behaviors among vets, due to a lack of testing facilities and their high cost [37,50–52].

While farm biosecurity and herd management are considered promising strategies to reduce the need for antimicrobials and ensure animal welfare, health, and production, improper farm management may lead to increased use of antimicrobials [53,54]. Consistent with previous studies [55,56], vets in this study stated that due to poor biosecurity in farms, they have to prescribe antimicrobials because of increased disease incidence, and this could be reduced if farms maintained biosecurity appropriately. Moreover, the farming system is another factor mentioned by the participants in the study. Evidence suggests that intensive

farming systems require higher input of antimicrobials resulting in higher rates of AMR in livestock production compared to traditional farming systems [6].

The impact of climate change on weather was found to be another factor that prompts vets to prescribe antimicrobials. Due to increased temperature and humidity, there is a higher risk for animals to be infected by diseases (flu or viral) [57]. Rodríguez-Verdugo and colleagues showed that local and global changes in temperature are associated with increases in antibiotic resistance and its spread [58]. Other researchers have suggested that increased temperatures, frequency, and intensity of heat waves may affect livestock health by causing metabolic disruptions, oxidative stress, and immune suppression which can lead to increased reliance on antimicrobials [59]. Consistent with this recently evolving evidence, participants in this study also stated that their antimicrobial prescriptions were also aimed at tackling heat induced diseases.

Regarding social opportunities, this study revealed that veterinarians' prescribing behaviors are influenced by different social actors including senior colleagues, peers, farmers, and some informal stakeholders (e.g., veterinary drug and feed sellers, pharmaceutical companies and the owners of the farm). In line with earlier studies [60,61], peers and senior colleagues play a vital role in prescribing antimicrobials through personal communication, professional groups, and different forums. As influential professional group members, peers also act as a catalyst in changing prescription behaviors of veterinarians and promoting reduced antimicrobials prescribing [62].

Farmers' or owners' desire to obtain antimicrobials for their animals is a well-known factor of prescribing antimicrobials in the international literature [60,63–66]. This may be due to the community perceptions of antimicrobials (such as antimicrobials are 'super medicine' that can solve any diseases) in curing diseases rapidly. Participants in this study stated that farmers and owners associate a 'good veterinarian' with their prescribing of antibiotics. As a result, farmers do not seek services from those veterinarians who do not prescribe antibiotics. Instead, they end up consulting informal quacks who are not authorized to prescribe antimicrobials. Additionally, vets in this study expressed fear of losing their clients. These findings are in line with earlier studies conducted in Bangladesh, India, Australia [17,43,52,67].

Veterinarians also are influenced to prescribe by some indirect stakeholders such as agents of veterinarian pharmaceutical companies and drug and feed sellers [68]. Usually, vets come to know about new groups or brands of antimicrobials from these stakeholders. They visit veterinarians' offices and introduce new groups and brands of medicines and feed and ask the veterinarians to prescribe them. Sometimes, veterinarians are influenced by their promotional campaigns and prescribe the promoted antibiotics.

In line with COM-B constructs and TDF domains, the qualitative findings of this study revealed a host of factors that motivate veterinarians both positively and negatively. Positive motivating factors included their role in AMS, professional identity, and social position. Negative motivating factors included their beliefs about consequences of reduced antimicrobial prescribing, pessimism, and inability to appropriately administer antimicrobials.

In Bangladesh, veterinarians are the only professionals who have the legal authority to prescribe antimicrobials to animals [69]. This is the strongest motivation for veterinarians to best use their knowledge and skills and change their prescribing behaviors. However, they alone cannot promote reductions in antimicrobial use because of the negative beliefs around the consequences of reducing antimicrobial prescribing, poor veterinary infrastructure, limited physical opportunities, complexity of the practice and the lack of monitoring of illegal sales by informal stakeholders. Additionally, if they do not prescribe, farmers will go to informal stakeholders, such as drug and feed sellers, who suggest or give antimicrobials without following any scientifically approved method [68,70,71]. Hence, it appears that reducing the use of antimicrobials is a shared responsibility for all stakeholders taking part in this sector.

## 6. Study limitation and strengths

This study was undertaken with a small sample from thousands of veterinarians who are operating in Bangladesh. The sample size was determined based on the qualitative methodological aspects such as saturation, participants' field experience, geographical coverage, and type of affiliations. Another limitation was that we included participants who are providing veterinary services to farm animals only. That means, we were not able to get insights from veterinarians who treat other animals such as companion animals, (or the pets) and domestic animals. However, since the overuse of antimicrobials is higher in farm animals, our objective was to understand the factors influencing antimicrobial prescription behavior of the population of veterinarians who are dealing with farm animals. Moreover, our findings may not be generalizable to this sector and not applicable for other countries; rather, they are subject to context-based analysis and reference. In other words, achieving generalizability of the findings was not our purpose, as we focused on capturing contextual factors behind prescribing antimicrobials by applying thematic analysis in order to relate our findings with existing literature in Bangladesh, other LMICs, and elsewhere around the world.

The COM-B model is an established method for understanding behavior and is used extensively in behavior change interventions in health, nutrition, gender, and agriculture domains across developed and developing countries. To the best of our knowledge, this is the first study that has applied a COM-B model in exploring the factors behind veterinarians' antimicrobial prescribing behaviors in LMICs. Using the COM-B model as the guiding principle for this study, our findings on the relevant domains of TDF allowed us to understand the nuances of factors influencing antimicrobial prescription behavior by registered veterinarians in Bangladesh. Use of the COM-B model and TDF domains increased our study's validity and credibility of the identified factors.

## 7. Recommendations

Based on our findings, we have several recommendations regarding policy changes and the design of future AMS interventions in Bangladesh. At the policy level, based on the experiential narratives by the vets in this study, it is critical to provide facilities for cost effective and timely antibiotic susceptibility testing, to encourage veterinarians to perform these tests, and to encourage farmers to bring their animals for testing. As a policy level intervention, the government should invest more in building an enabling environment so vets can receive antibiotic susceptibility testing results in a timely manner and encouraging them to make evidence-based decisions in antimicrobial prescribing. Acknowledging the shared responsibility among stakeholders in reducing antimicrobial use, collaborative efforts should be encouraged. This involves engaging not only veterinarians but also farmers, regulatory bodies, and other relevant entities to collectively address the complex challenges associated with antimicrobial use. The government should include all these stakeholders and work with them to address this shared challenge.

We also recommend that future AMS interventions should use behavior change wheel as the guiding tool in determining what types of interventions may be effective in combatting the inappropriate use of antimicrobials in the field. For example, continued education or training, providing up-to-date information and skill building workshops or seminars could be relevant interventions for addressing factors related to capability (of COM-B model) and knowledge, skills, abilities, and proficiencies (under TDF domains). In order to reduce the disease burden, providing adequate guidance to farmers for maintaining biosecurity on farms, vaccinating their animals, and educating different social actors such as peers, farmers or owners, and informal stakeholders about their role in reducing the AMR burden are some approaches to

consider. Educating farmers and other informal stakeholders about unnecessary antimicrobial use could reduce the demand for improper antibiotics. Finally, veterinarians should focus on building strong relationships with farmers to reduce farmers' likelihood to seek antimicrobials elsewhere, particularly from informal quacks.

## Supporting information

**S1 File. Supplementary on COREQ implementation.**
(DOCX)

## Acknowledgments

We would like to thank the research participants who gave their valuable time and participated in the study voluntarily. We also acknowledge the support from Chattogram Veterinary and Animal Science University in granting ethical approval of this study. We also acknowledge the contribution of Irina Bergenfeld (a PhD Student in Global Health and Development Program, Emory University, Atlanta, Georgia, USA) for her time and effort to proofread the manuscript. The authors acknowledge with particular gratitude the anonymous reviewers who offered detailed and helpful comments on the manuscript.

## Author Contributions

**Conceptualization:** Shahanaj Shano, Md Abul Kalam.

**Data curation:** Md. Sahidur Rahman, Samira Akter, Md Nasir Uddin, Faruk Ahmed Jalal.

**Formal analysis:** Shahanaj Shano, Md Abul Kalam, Sharmin Afrose, Samira Akter, Md Nasir Uddin, Faruk Ahmed Jalal.

**Funding acquisition:** Mohammad Mahmudul Hassan.

**Investigation:** Md. Sahidur Rahman, Samira Akter, Md Nasir Uddin, Faruk Ahmed Jalal.

**Methodology:** Shahanaj Shano, Md Abul Kalam.

**Project administration:** Shahanaj Shano, Md Abul Kalam, Md. Sahidur Rahman, Md Nasir Uddin.

**Resources:** Mohammad Mahmudul Hassan.

**Software:** Shahanaj Shano, Md Abul Kalam, Sharmin Afrose.

**Supervision:** Maya L. Nadimpalli.

**Validation:** Shahanaj Shano, Md Abul Kalam, Sharmin Afrose, Maya L. Nadimpalli.

**Visualization:** Shahanaj Shano, Md Abul Kalam.

**Writing – original draft:** Shahanaj Shano.

**Writing – review & editing:** Md Abul Kalam, Sharmin Afrose, Pronesh Dutta, Mithila Ahmed, Khnd Md Mostafa Kamal, Mohammad Mahmudul Hassan, Maya L. Nadimpalli.

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
