## [Decision Letter · Decision Letter 0]

21 Nov 2023

PONE-D-23-31043An Application of COM-B Model to Explore Factors Influencing Veterinarians’ Antimicrobial Prescription Behaviors: Findings from a Qualitative Study in Bangladesh.PLOS ONE

Dear Dr. Kalam,

Thank you for submitting your manuscript to PLOS ONE. After careful consideration, we feel that it has merit but does not fully meet PLOS ONE’s publication criteria as it currently stands. Therefore, we invite you to submit a revised version of the manuscript that addresses the points raised during the review process.

The reviewers saw great value in your work which will be enhanced with careful consideration of the reviewers comments

We look forward to receiving your revised manuscript.

Kind regards,

Lian Francesca Thomas, PhD

Academic Editor

PLOS ONE

Journal Requirements:

2. Thank you for submitting the above manuscript to PLOS ONE. During our internal evaluation of the manuscript, we found significant text overlap between your submission and previous work in the [introduction, conclusion, etc.].

Please revise the manuscript to rephrase the duplicated text, cite your sources, and provide details as to how the current manuscript advances on previous work. Please note that further consideration is dependent on the submission of a manuscript that addresses these concerns about the overlap in text with published work.

[If the overlap is with the authors’ own works: Moreover, upon submission, authors must confirm that the manuscript, or any related manuscript, is not currently under consideration or accepted elsewhere. If related work has been submitted to PLOS ONE or elsewhere, authors must include a copy with the submitted article. Reviewers will be asked to comment on the overlap between related submissions (http://journals.plos.org/plosone/s/submission-guidelines#loc-related-manuscripts).]

We will carefully review your manuscript upon resubmission and further consideration of the manuscript is dependent on the text overlap being addressed in full. Please ensure that your revision is thorough as failure to address the concerns to our satisfaction may result in your submission not being considered further.

"We would like to thank the research participants who gave their valuable time and took participate in the study voluntarily. We also acknowledge the support from Chattogram Veterinary and Animal Science University in granting ethical approval of this study. We also grateful to Bangladesh Bureau of Educational Information and Statistics (BANBEIS) for providing partial funding (ID#SD2019967) to this study. We also acknowledge the contribution of Irina Bergenfeld (a PhD Student in Global Health and Development Program, Emory University, Atlanta, Georgia, USA) for her time and effort to proofread the manuscript. The authors acknowledge with particular gratitude the anonymous reviewers who offered detailed and helpful comments on the manuscript."

"This work was funded by Bangladesh Bureau of Educational Information and Statistics (BANBEIS), ID number SD2019967. Mohammad Mahmudul Hassan was supported by BANBEIS. The funders had no role in study design, data collection and analysis, decision to publish, or preparation of the manuscript."

"NO authors have competing interests."

Reviewers' comments:

Reviewer's Responses to Questions

**Comments to the Author**

1. Is the manuscript technically sound, and do the data support the conclusions?

Reviewer #1: Yes

Reviewer #2: Partly

2. Has the statistical analysis been performed appropriately and rigorously? 

Reviewer #1: N/A

Reviewer #2: N/A

3. Have the authors made all data underlying the findings in their manuscript fully available?

Reviewer #1: No

Reviewer #2: No

4. Is the manuscript presented in an intelligible fashion and written in standard English?

Reviewer #1: Yes

Reviewer #2: Yes

5. Review Comments to the Author

Reviewer #1: Dear Author(s)/Editor(s),

This is a well-written paper with interesting findings. Antimicrobial stewardship is indeed an essential strategy in addressing AMR and veterinarians play a key role in this endeavor. To this end, the application of comprehensive and theory driven behavioral science frameworks such as the COM-B model are both welcomed and necessary. Application of these approaches is important as it highlights factors beyond “knowledge deficits” that pattern antimicrobial use patterns and strives for a more holistic approach beyond the common training strategies focused on awareness raising. To this end, the authors should be commended for taking a behavioral science approach towards understanding veterinary prescription factors. As they correctly note, veterinarians play a key role in determining the types of antimicrobials used and whether these antimicrobials are correctly administered. This is especially the case in LMICs, as regulatory frameworks and guidance are limited, resulting in veterinarians drawing upon on their own experiences to dispense antimicrobials and often in competition with “quacks”, who may not have the background to prescribe the correct antimicrobial. Given the essential role played by veterinarians in animal health, and by extension AMR, the results presented have the potential to make a significant contribution to the field. With this said, we suggest a few major and several minor revisions that should be considered before publication. We also attach a pdf with minor comments.

Major Revisions

We would suggest that the findings be clustered more carefully into the behavioral science frameworks. There were a few instances where it was difficult to see how certain codes were clustered into a particular domain, especially when there was ambiguity. A stronger justification for these clustering decisions would be helpful (specific instances with corresponding line numbers found in “Minor Revisions”).

The recommendations focus heavily on educational interventions, with relatively little attention given to interventions addressing motivation. It would be helpful to provide a more balanced approach to the recommendations, given that motivation is a critical factor in the COM-B.

The data analysis section could be clarified to better explain how the codes were clustered into the frameworks and how COM-B was combined with the TDF. Providing more specific examples to illustrate your points would also be helpful. (Again, specific instances with corresponding line numbers found in “Minor Revisions”)

Finally, the paper does not investigate cultural and psychological factors in depth. These factors can have a significant impact on behavior, and it would be helpful to consider them more fully in the analysis and discussion or possibly as a recommendation for Future Research.

Minor revisions (Line Number preceding comment)

36: Consider rephrasing the term "behavioral factors" throughout the document, as it is not technically accurate. The study investigates the factors that influence behavior, many of which are not behavioral in nature (e.g., opportunities)

39: Clarify gap: There is a large body of research on the factors that influence veterinarian behavior, however, there is a need for more studies that use comprehensive behavior change models to develop and evaluate interventions. This is the value this study adds from my perspective.

107 – 110. There is a growing body of evidence that suggests that the constructs used in these models are not always reliable predictors of behavior. This is known as the intention-action gap, which is a well-studied phenomenon (including agricultural sciences). As a result, the theories mentioned here are no longer recommended.

122-123: It is important to note that the COM-B model and the Behavior Change Wheel (BCW) are two different but complementary frameworks. While the COM-B model is specifically designed to investigate the factors that influence behavior, the BCW is a framework for identifying and selecting behavior change interventions. The authors may want to consider focusing on the COM-B model in the main body of the document, and mentioning the BCW in the discussions when recommendations for interventions are made. This is because the COM-B model is a more general framework for understanding the three essential components of behavior, while the BCW is a more specific framework for identifying and selecting interventions.

125: This is how the TDF was created. The COM-B is based on 1) a US consensus meeting of behavioural theorists in 1991 and 2) principle of US criminal law.

130: The authors may want to consider using the original definition of motivation (and the other COM-B domains) from the original paper. This definition is more inclusive of both reflective and automatic decision-making.

133: The authors may want to clarify that the TDF is not built upon the COM-B model. In fact, the first version of the TDF was published in 2005, while the COM-B model was published in 2011 (both Michie et al.). However, the two frameworks are complementary, and the revised version of the TDF by Cane et al. (2012) shows how they can be combined to advance the level of detail and complexity.

133-136: The last two sentence could be simplified for readers not familiar with this.

146: Also consider these studies from same authors:

- Farrell, S., Benson, T., McKernan, C., Regan, Á., Burrell, A. M., & Dean, M. (2023). Factors influencing dairy farmers' antibiotic use: An application of the COM-B model. Journal of Dairy Science, 106(6), 4059-4071.

- Farrell, S., Benson, T., McKernan, C., Regan, Á., Burrell, A., & Dean, M. Exploring Veterinarians’ Behavior Relating to Antibiotic Use on Dairy Farms Using the COM-B Model of Behaviour Change. Available at SSRN 4328183.

146: Additional review to consider: Biesheuvel, M. M., Santman-Berends, I. M., Barkema, H. W., Ritter, C., Berezowski, J., Guelbenzu, M., & Kaler, J. (2021). Understanding farmers' behavior and their decision-making process in the context of cattle diseases: a review of theories and approaches. Frontiers in veterinary science, 8, 687699.

214: It would be helpful to clarify the distinct steps involved in developing the deductive and inductive codes.

217: The authors should clarify if and how the TDF was used as an analytical framework in the paper. It is mentioned throughout, however, not mentioned here. Additionally, it is unclear how the different types of codes generated are used.

239: Add reference (Cane et al. (2012)). Please clarify how the TDF was used as an analytical framework in the paper, or if it wasn’t used remove the TDF column. If the TDF was used, please specify which inductive codes were mapped to the TDF domains.

273: self-efficacy. I suspect it is “perceived competence” but this needs a quote to conform. Additionally, the concept is not explicitly mentioned in the table above, but it could be classified under the TDF domain of "beliefs about capabilities" and the COM-B component of "reflective motivation."

352: It may be helpful to clarify that the statement "participants believe that livestock owners focus on money and want to raise more poultry per land area to minimize costs while maximizing profit" represents the vets' beliefs, not necessarily reality.

376: Avoid mentioning numbers in a qualitative study, unless they are essential to the findings or to provide context.

436: Clarify if this is what participants believe

460: Fear of losing job. Consider this moving to motivation either as emotion or as beliefe of consequences.

467: …to promote reduced AMU/act as AMR stewards. Use the same behavior you are looking at throughout the paper.

493: Please explain why the component described in this section is classified as "motivation," since this component describes internal processes that trigger or inhibit behaviour.

520: Beliefs about impact. TDF term is beliefs about consequences

535: Quote suggest that the fear of losing one's reputation due to unsuccessful treatment is the underlying driver of this behavior

560 – 564: Behaviors influenced by the actions or perceptions of others should be classified under the "social opportunity" domain.

565-567: Please explain how lack of monitoring or law enforcement can influence motivation. It may also fall under external opportunities depending on the exact quote.

569-570: elaborate why this is important or relevant.

575: Inductive inference. Please consider if this is more appropriately conceived as deductive analysis

609: consider rephrasing to “reduce the need for AMs

674: The data analysis section does not mention thematic analysis. Also mention here, that both thematic analysis and framework analysis was used.

681: Please explain how the two frameworks were integrated into the data analysis

699-701: I am not sure how the BCW can directly suggest this. A better approach might be to start with the broader policy (or intervention function) categories and then move on to specific suggestions

Reviewer #2: The idea behind the study is great and presents useful information on the factors influencing antimicrobial prescription behaviours of Vets. However, the authors have not done justice to the paper especially given the nature of the framing of the paper especially at the presentation of results and subsequently the discussion. There is need to properly link the findings to the results from the informants and to the components of the COM-B Model

There is need to check the whole document for grammatical errors especially the quotes which need to be edited for grammar and sentence structure.

Line 81-83- Is the problem of misuse and abuse linked to prescription by vets or self-medication?

RESULTS

Summary 1 Table 1 page 13. How was knowledge defined as fair or lacking?

Specific comments include: There is need to indicate how knowledge was measured and what entailed fair knowlege.

Was knowledge equated with awareness?

Line 276- It is not clear whether the quote from line 277-282 is from an experienced or junior vet.

It is not clear whether the motivation is that vets are fearful about failure of treatment. How does this point link to influencing the Vets behaviour? There needs to be a clear linkage of this motivation

Line 301 seems to be hanging

Under sub topic 4.2.2 Line 306 – 307 is framed more like a barrier as opposed to an opportunity…. Generally the framing of the results need to bring out the key components under the COM-B Model eg the opportunities sound more like barriers, it would be great to frame them as opportunities that vets take advantage of to prescribe antibiotics

It is not clear whether the opportunities influences negatively or positively , This needs to come out clearly – How is environmental factors an opportunity and is it an opportunity for antimicrobial prescription behaviour of a farmer?

Line 322-328: As a result, farmers 324 feel reluctant to bring their large animals (for example ruminants or goats) due to the transportation325 hassle and cost. One participant said,

326 “……this is particularly a huge barrier for the farmers those who have cows or goats due

327 to the long distance and cost. Because farmers have to hire a transportation like motorized

328 van which costs a lot of money.” (Participant – 12). The authors have not linked this well with the opportunity for vets to reduce prescription of antimicrobials.

Under line 331-335 – Do the authors mean animal management of is it farm management? From my perspective farm management is about the infrastructure of the farm.

Authors need to clearly relate the aspects of the facilities related to farm management and biosecurity to veterinarians’ behaviour in relation to antimicrobial prescription.

Its important to bring out the voices of the informants. In the text it is sometimes not easy to differentiate the voices of the authors from those of the informants eg

Line 467 sub title: 4.2.3 Veterinarians’ motivation to promote reduced AMU/act as AMR stewards

The text under this sub title indicate what the vets would love to see but the authors could consider outlining and presenting information in relation to what t motivates them to promote reduced AMY/act as Stewards

In some situations the authors talk about farmer behaviour. It is not clear how the farmers behaviour is linked to the vets behaviour in relation to prescribing antimicrobials.

How have the socio-demographic aspects of the informants been weaved into the presentation of the results?

6. PLOS authors have the option to publish the peer review history of their article (what does this mean?). If published, this will include your full peer review and any attached files.

Reviewer #1: No

Reviewer #2: No

---

## [Author Response · Author response to Decision Letter 0]

16 Feb 2024

Response: We have adhered to PLOS’s style requirements. Please check in the revised version. 

2. Thank you for submitting the above manuscript to PLOS ONE. During our internal evaluation of the manuscript, we found significant text overlap between your submission and previous work in the [introduction, conclusion, etc.].

Please revise the manuscript to rephrase the duplicated text, cite your sources, and provide details as to how the current manuscript advances on previous work. Please note that further consideration is dependent on the submission of a manuscript that addresses these concerns about the overlap in text with published work.

[If the overlap is with the authors’ own works: Moreover, upon submission, authors must confirm that the manuscript, or any related manuscript, is not currently under consideration or accepted elsewhere. If related work has been submitted to PLOS ONE or elsewhere, authors must include a copy with the submitted article. Reviewers will be asked to comment on the overlap between related submissions (http://journals.plos.org/plosone/s/submission-guidelines#loc-related-manuscripts).]

We will carefully review your manuscript upon resubmission and further consideration of the manuscript is dependent on the text overlap being addressed in full. Please ensure that your revision is thorough as failure to address the concerns to our satisfaction may result in your submission not being considered further.

Response: This comment was withdrawn by the journal on February 09 via email. However, we have revised the introduction and recommendation sections, although these modifications do not substantially alter the narrative of the manuscript.

"We would like to thank the research participants who gave their valuable time and took participate in the study voluntarily. We also acknowledge the support from Chattogram Veterinary and Animal Science University in granting ethical approval of this study. We also grateful to Bangladesh Bureau of Educational Information and Statistics (BANBEIS) for providing partial funding (ID#SD2019967) to this study. We also acknowledge the contribution of Irina Bergenfeld (a PhD Student in Global Health and Development Program, Emory University, Atlanta, Georgia, USA) for her time and effort to proofread the manuscript. The authors acknowledge with particular gratitude the anonymous reviewers who offered detailed and helpful comments on the manuscript."

"This work was funded by Bangladesh Bureau of Educational Information and Statistics (BANBEIS), ID number SD2019967. Mohammad Mahmudul Hassan was supported by BANBEIS. The funders had no role in study design, data collection and analysis, decision to publish, or preparation of the manuscript."

Response: We have dropped out the funding information from the acknowledgement section of the revised manuscript (Please check line number 770-780 in the revised manuscript). Please see the funding statement below: 

"This work was funded by Bangladesh Bureau of Educational Information and Statistics (BANBEIS), ID number SD2019967. The funders had no role in study design, data collection and analysis, decision to publish, or preparation of the manuscript." 

"NO authors have competing interests."

Response: Please see our Competing Interests below: 

Response: Please find our updated data availability statement below: 

Dataset is not publicly shareable because the participants of the study did not give consent to make their interview transcripts available in the public domain. Additionally, the transcript may contain some sensitive information such as the brand name of the antimicrobials, as well as participants could be identifiable because of their professional affiliation, which may put them at risk. However, data request can be made to the corresponding author at mkalam@emory.edu or to the ethical committee directly at drecvasu@gmail.com. An anonymized dataset can be available with an appropriate purpose. 

Response: We have added Supporting Information file at the end of the manuscript. We also cited this document within the text. 

Reviewers' comments:

Reviewer's Responses to Questions

Comments to the Author

1. Is the manuscript technically sound, and do the data support the conclusions?

Reviewer #1: Yes

Reviewer #2: Partly

2. Has the statistical analysis been performed appropriately and rigorously?

Reviewer #1: N/A

Reviewer #2: N/A

3. Have the authors made all data underlying the findings in their manuscript fully available?

Reviewer #1: No

Reviewer #2: No

4. Is the manuscript presented in an intelligible fashion and written in standard English?

Reviewer #1: Yes

Reviewer #2: Yes

5. Review Comments to the Author

Reviewer #1: Dear Author(s)/Editor(s), 

Comment: This is a well-written paper with interesting findings. Antimicrobial stewardship is indeed an essential strategy in addressing AMR and veterinarians play a key role in this endeavor. To this end, the application of comprehensive and theory driven behavioral science frameworks such as the COM-B model are both welcomed and necessary. Application of these approaches is important as it highlights factors beyond “knowledge deficits” that pattern antimicrobial use patterns and strives for a more holistic approach beyond the common training strategies focused on awareness raising. To this end, the authors should be commended for taking a behavioral science approach towards understanding veterinary prescription factors. As they correctly note, veterinarians play a key role in determining the types of antimicrobials used and whether these antimicrobials are correctly administered. This is especially the case in LMICs, as regulatory frameworks and guidance are limited, resulting in veterinarians drawing upon on their own experiences to dispense antimicrobials and often in competition with “quacks”, who may not have the background to prescribe the correct antimicrobial. Given the essential role played by veterinarians in animal health, and by extension AMR, the results presented have the potential to make a significant contribution to the field. With this said, we suggest a few major and several minor revisions that should be considered before publication. We also attach a pdf with minor comments.

Response: Thanks so much for your time and enthusiasm to read the manuscript and provided valuable feedback. 

Major Revisions

Comment: We would suggest that the findings be clustered more carefully into the behavioral science frameworks. There were a few instances where it was difficult to see how certain codes were clustered into a particular domain, especially when there was ambiguity. A stronger justification for these clustering decisions would be helpful (specific instances with corresponding line numbers found in “Minor Revisions”).

Response: Thanks so much for your comment. As we mentioned in the draft, we used the COM-B model as our guiding principle to carry out our study and our analysis was built on both inductive and deductive coding. Hence, the decision for clustering was done by matching the text segments and the nature of the inductive codes. Before making the final decision on the case of ambiguity, we took two strategies. Firstly, consulting with the literature and secondly, consensus among the coders. It is important to note here, since there is lack of studies available on the use of COM-B model in studying the behavioral determinants of antimicrobial prescription by the veterinarians, we consulted with other research areas such as prescription of antibiotics (and other medicines) for human use, and other health issues. To achieve the consensus among coders, the coders discussed among themselves and made the final decision. Please see related information in line number 250-260 in the revised paper. 

Comment: The recommendations focus heavily on educational interventions, with relatively little attention given to interventions addressing motivation. It would be helpful to provide a more balanced approach to the recommendations, given that motivation is a critical factor in the COM-B.

Response: Thanks so much for your comment. We have now added text on other areas of recommendations including strategies to increase motivation. Please see the revised version throughout which starts with policy priority to make the testing facility available first, followed by increase the sense of shared responsibilities through collective efforts and increase the motivation and other domains of the intervention based on the findings of different domains of COM-B model. 

Comment: The data analysis section could be clarified to better explain how the codes were clustered into the frameworks and how COM-B was combined with the TDF. Providing more specific examples to illustrate your points would also be helpful. (Again, specific instances with corresponding line numbers found in “Minor Revisions”)

Response: Thanks so much for your comment. As we mentioned in our earlier response, we first developed the deductive codes based on the domains of the COM-B model and the data collection guideline. We developed the inductive codes based on what the research participants responded. Then these codes were categorized and clustered based on the attributes, (definitions and applicability) of the codes and the domains of the COM-B model. This approach has been described in the book titled “Qualitative Research Methods” – by Hennink, Hutter and Bailey. However, the combination of COM-B model and the TDF was made based on the similarities of each domain. In doing so, we grouped the sub-domains of TDF under different domains of COM-B model, as done by Annemarie De Leo and colleagues in their 2021 study (PMID 33436092). De Leo et al. was referred and cited in the paper (Figure 2 in that paper for your reference). Table 2 in the original draft has presented this strategy. 

Comment: Finally, the paper does not investigate cultural and psychological factors in depth. These factors can have a significant impact on behavior, and it would be helpful to consider them more fully in the analysis and discussion or possibly as a recommendation for Future Research.

Response: Thanks so much for your observation. For us, the domain “Opportunity” broadly covers cultural factors. Especially, the sub-domain titled “Social Influence” captured many of the aspects of cultural factors of antimicrobial prescriptions. As the original article by Michie et al mentioned “the environment and social opportunity afforded by the cultural milieu that dictates the way that we think about things (e.g., the words and concepts that make up our language).” [PMID: 21513547 or https://doi.org/10.1186/1748-5908-6-42] However, we do agree that the socio-cultural properties around antimicrobials were not captured in-depth, in part due to the adopting the COM-B framework. 

However, for the psychological factors, our findings on psyc

---

## [Editor Report · Decision Letter 1]

17 Mar 2024

PONE-D-23-31043R1An Application of COM-B Model to Explore Factors Influencing Veterinarians’ Antimicrobial Prescription Behaviors: Findings from a Qualitative Study in Bangladesh.PLOS ONE

Dear Dr. Kalam,

Thank you for submitting your manuscript to PLOS ONE. After careful consideration, we feel that it has merit but does not fully meet PLOS ONE’s publication criteria as it currently stands. Therefore, we invite you to submit a revised version of the manuscript that addresses the points raised during the review process. 

Thank you for revising the manuscript.

As new editor assigned to the document I read through the document.

Some comments:

It looks like the concept of selection of resistance by the participants is not always known (L297), in next paragraph some other participants correctly use the concept of selection of resistance, though the example given states antibiotic resistance will be created in the human body, which indicates that this participant does not know how resistance evolves. L308 speaks again over development of AMR, while it should be selection of AMR. Discuss this and use correct terminology.

Participant 14, L328, shows another problem and that is that penicillin is not effective against the bacteria listed… The participant does not know what antibiotics to use for which infection

This indicates that there are other gaps in the knowledge. Please address

L734 Companion animals and pets are the same, they are moreover domestic, adapt

The comments of the reviewers have been appropriately replied to

We look forward to receiving your revised manuscript.

Kind regards,

Patrick Butaye, DVM, PhD

Academic Editor

PLOS ONE
---

## [Author Response · Author response to Decision Letter 1]

19 Apr 2024

Comment: It looks like the concept of selection of resistance by the participants is not always known (L297), in next paragraph some other participants correctly use the concept of selection of resistance, though the example given states antibiotic resistance will be created in the human body, which indicates that this participant does not know how resistance evolves. 

Response: Thanks for your comment and identify the nuance. However, given the different pathways of growing AMR, our participants are aware how AMR evolves and provided these two explanations. Regarding to the spread of AMR to human, that specific participant meant how consumption of chicken may lead to human AMR. This is an example. That means, this another way to grow AMR. This is an awareness related both animal and human AMR. However, we have edited the texts to increase the clarity. Please check line number 248-253 in the revised version. 

Comment: L308 speaks again over development of AMR, while it should be selection of AMR. Discuss this and use correct terminology.

Response: Thanks so much for your comment. This section is about participants’ knowledge on AMR and how it develops. However, in the consecutive sub-sections of the result section, we have outlined the other drivers that influence the selection of antimicrobials. We have edited the language for the clarity. Please check line number 263 to 264 in the revised version. 

Comment: Participant 14, L328, shows another problem and that is that penicillin is not effective against the bacteria listed… The participant does not know what antibiotics to use for which infection. 

This indicates that there are other gaps in the knowledge. Please address

Response: Thanks so much for pointing this out. We were looking back at our transcript. We think there was a typo during the translation phase. Basically, the participant mentioned “Ciprofloxacin” which is now edited and updated. Please check the line number 282 in the revised version. 

Comment: L734 Companion animals and pets are the same, they are moreover domestic, adapt

Response: Thanks for your catching this. We have edited in the revised version. Please check the line number 677 in the revised version.

---

## [Decision Letter · Decision Letter 2]

6 Nov 2024

PONE-D-23-31043R2An Application of COM-B Model to Explore Factors Influencing Veterinarians’ Antimicrobial Prescription Behaviors: Findings from a Qualitative Study in Bangladesh.PLOS ONE

Dear Dr. Kalam,

Thank you for submitting your manuscript to PLOS ONE. After careful consideration, we feel that it has merit but does not fully meet PLOS ONE’s publication criteria as it currently stands. Therefore, we invite you to submit a revised version of the manuscript that addresses the points raised during the review process.

We look forward to receiving your revised manuscript.

Kind regards,

Muhammad Ahmad

Academic Editor

PLOS ONE

Journal Requirements:

Additional Editor Comments:

revision required

Reviewers' comments:

Reviewer's Responses to Questions

**Comments to the Author**

1. If the authors have adequately addressed your comments raised in a previous round of review and you feel that this manuscript is now acceptable for publication, you may indicate that here to bypass the “Comments to the Author” section, enter your conflict of interest statement in the “Confidential to Editor” section, and submit your "Accept" recommendation.

Reviewer #3: All comments have been addressed

Reviewer #4: (No Response)

Reviewer #5: All comments have been addressed

Reviewer #6: All comments have been addressed

Reviewer #7: All comments have been addressed

Reviewer #8: (No Response)

Reviewer #9: All comments have been addressed

Reviewer #10: (No Response)

2. Is the manuscript technically sound, and do the data support the conclusions?

Reviewer #3: Yes

Reviewer #4: Yes

Reviewer #5: Yes

Reviewer #6: Yes

Reviewer #7: Partly

Reviewer #8: Yes

Reviewer #9: Yes

Reviewer #10: Yes

3. Has the statistical analysis been performed appropriately and rigorously? 

Reviewer #3: Yes

Reviewer #4: I Don't Know

Reviewer #5: N/A

Reviewer #6: I Don't Know

Reviewer #7: N/A

Reviewer #8: No

Reviewer #9: Yes

Reviewer #10: I Don't Know

4. Have the authors made all data underlying the findings in their manuscript fully available?

Reviewer #3: Yes

Reviewer #4: No

Reviewer #5: No

Reviewer #6: Yes

Reviewer #7: Yes

Reviewer #8: No

Reviewer #9: Yes

Reviewer #10: Yes

5. Is the manuscript presented in an intelligible fashion and written in standard English?

Reviewer #3: Yes

Reviewer #4: Yes

Reviewer #5: Yes

Reviewer #6: Yes

Reviewer #7: Yes

Reviewer #8: Yes

Reviewer #9: Yes

Reviewer #10: Yes

6. Review Comments to the Author

Reviewer #3: Comments raised by the reviewers has been addressed satisfactorily by the authors, the manuscript is technically and scientifically sound, the content can contribute immensely to the field of antimicrobial resistance.

Reviewer #4: The research article entitled “An Application of COM-B Model to Explore Factors Influencing Veterinarians’ Antimicrobial Prescription Behaviors: Findings from a Qualitative Study in Bangladesh” has been reviewed. In the study, Shano and co-workers applied the Capability, Opportunity and Motivation for Behaviour (COM-B) model to comprehensively study the behavioural change models to evaluate antimicrobial prescriptive interventions among registered veterinarians in Bangladesh. The study mainly involved one-on-one interviews with 32 participants from the aforementioned professionals. Positively, it could have impacted the policy makers on necessary regulations on the prescription behaviours of antimicrobial prescriptions by veterinarians in Bangladesh and some other nations affected by the knowledge covered. However, most of the highlighted outcomes are ideally expected to have been undertaken by the professionals as important parts of curriculum during the course of training. If so, then what is the significance of this study? Authors need to strongly indicate that the new findings are not already existing knowledge in the course of training professional veterinarians that prescribe antimicrobials or channel their efforts towards the implementation of the known knowledge/policy. Based on this and other comments enlisted below, I recommend a major revision before the manuscript can be accepted for publication in this title.

1. Understandably, the cause of antimicrobial resistance extends beyond an excessive usage but also importantly entails misuse of antimicrobials and several other factors including poor hygiene, lack of sanitation, contamination of food, drink and environments, inadequate preventive and control measures etc. How have the authors considered the influence of these other factors on the prescription behaviour of veterinarians? In addition, evolvement of resistance mechanisms towards the existing medications oftentimes constitute an increase in antimicrobials prescription. Authors need to also consider this in their recommendations.

2. Were the COM-B practices not already in the curriculum or courses of studies undertaken by the formal professionals during their trainings? If yes, what is the significance of this present study? Perhaps the results from this study could discourage

3. Authors are advised to minimize the usage of personalized statements and instead, adopt reporting formats. E.g. use “the study was conducted by…” instead of “we conducted the study by…”

4. What influenced the significant gender bias in authors’ criteria for participant selections and how could these not affect the fairness in the outcomes?

5. In section 3.2, the authors have mentioned bachelor degree holders among the recruited participants. However, this class is missing in the table of participant’s characteristics (Table 1). Kindly clarify.

6. I strongly advise that the knowledge depths of the selected participants should also be considered to validate the results from this study. For instance, if a farmer intends to sell his chickens on 30th and they developed microbial infection on 27th as mentioned by a participant, what should be done by the farmer for his business security and human safety?

7. Were the quantitative data from this study analyzed statistically? Kindly indicate the location if available.

8. The English language of this submission requires a thorough recheck by an expert. Authors should consider this.

Reviewer #5: (No Response)

Reviewer #6: The previous comments have been addressed. However in responses, Line 281 it is mentioned that ciprofloxacin is narrow spectrum antibiotic. Kindly check the transcript again or otherwise this still shows some knowledge gaps as ciprofloxacin is broad spectrum.

Reviewer #7: Both reviewers are well explained each of the statement. Its an interesting survey work. But one things, Its not a clear idea how antibiotic resistance created human body after eating poultry products?

Reviewer #8: Dear authors, I have the following concerns

1. The sample size (32) is too small to draw strong evidence to fulfil the objectives. according to the Bangladesh Veterinary Council, the country has more than eight thousand veterinarians, then how these 32 will present the other. This was also expressed under the limitation. However, it is too small to fulfil objectives.

2. Line 153, remove phrases like ----please see, put only something in the for further reference

3. Line 255, instead of saying “some of the participants” why don't you write the number and/or of participants who answered that question. This is true throughout the document including the abstract

4. Line 313 ------- Almost all the participants (n=24/32), this is 75%, not almost all... hence the sentence is not accurate

5. Keywords are absent in the main part of the manuscript just after the abstract. There is the list in the submission system ---------� Global Health, One Health, AMR, Infectious Disease, Antimicrobial Prescription, COMB Model, Theory of Domain Framework, Behavior Change Wheel, Qualitative, Thematic Analysis, Veterinarians, Veterinary Practices, Bangladesh.

a. The list is too much

b. Global Health, One Health, Infectious Disease, Theory of Domain Framework, Wheel, and Thematic Analysis are not described well in the manuscript, please cancel

c. Please don’t abbreviate the keywords, and separate by semi-colon

6. The conclusion was not presented separately and clearly

Reviewer #9: Thank you for the opportunity to review this manuscript. It presents a compelling and important study that explores the behavioural factors influencing antimicrobial prescription practices among registered veterinarians in Bangladesh. The authors have made a significant effort in addressing the feedback from both reviewers, and the manuscript demonstrates several strengths that warrant its acceptance for publication.

Here is a summary of the key reasons for my recommendation:

1. Relevance and impact: The topic of antimicrobial resistance and prescription behaviours among veterinarians is highly relevant, especially in low- and middle-income countries like Bangladesh. The manuscript addresses an urgent global health concern with a novel application of the COM-B model.

2. Robust theoretical framework: The authors have effectively used the COM-B model and Theoretical Domains Framework, which strengthens the manuscript’s contribution to the field of behaviour change in antimicrobial stewardship. The study is well-grounded in behavioural science, which adds depth to the findings.

3. Methodological rigor: The qualitative approach is sound, and the authors have provided clear explanations of their coding processes and thematic analysis. The use of both deductive and inductive methods ensures a rich dataset that aligns well with the research objectives.

4. Revisions based on reviewers’ feedback: The authors have responded appropriately to reviewers’ concerns. They have clarified terminology, corrected minor errors, and enhanced the manuscript’s clarity and precision. The response to the feedback shows a high level of engagement to improving the manuscript.

Conclusion:

With the effort the authors have put into addressing the reviewers' comments, the manuscript demonstrates sufficient rigor and relevance for publication. Given the timely and well-executed nature of the research, I would recommend acceptance, especially with the improvements made after the review process. The study's contribution to AMS practices in veterinary medicine, particularly in resource-limited settings, is valuable and adds to the growing body of global health literature.

Reviewer #10: L-42 semi-structured

L-63 as an invisible pandemic

L-70 lower-and-middle-income countries

L-71 as in the case of other LMICs

L-81 As in

L-82 the case of human antibiotics in Bangladesh

L-101 main behavioral factors that globally influence antimicrobials

L-114 behavior of veterinarians

L-121 We employed the Capability, Opportunity, Motivation and Behavior (COM-B)

L-126 The term ‘Capability’ refers to an

L-128 ‘Motivation’ denotes the brain

L-137 Conceptual framework: Integration of Theoretical Domains Framework (TDF) into the

7. PLOS authors have the option to publish the peer review history of their article (what does this mean?). If published, this will include your full peer review and any attached files.

Reviewer #3: **Yes: **Salim Faruk Bashir

Reviewer #4: No

Reviewer #5: No

Reviewer #6: No

Reviewer #7: No

Reviewer #8: No

Reviewer #9: **Yes: **Muhammad Augie Bashar

Reviewer #10: No

---

## [Author Response · Author response to Decision Letter 2]

19 Nov 2024

Comments to the Author

1. If the authors have adequately addressed your comments raised in a previous round of review and you feel that this manuscript is now acceptable for publication, you may indicate that here to bypass the “Comments to the Author” section, enter your conflict of interest statement in the “Confidential to Editor” section, and submit your "Accept" recommendation.

Reviewer #3: All comments have been addressed

Reviewer #4: (No Response)

Reviewer #5: All comments have been addressed

Reviewer #6: All comments have been addressed

Reviewer #7: All comments have been addressed

Reviewer #8: (No Response)

Reviewer #9: All comments have been addressed

Reviewer #10: (No Response)

2. Is the manuscript technically sound, and do the data support the conclusions?

Reviewer #3: Yes

Reviewer #4: Yes

Reviewer #5: Yes

Reviewer #6: Yes

Reviewer #7: Partly

Reviewer #8: Yes

Reviewer #9: Yes

Reviewer #10: Yes

3. Has the statistical analysis been performed appropriately and rigorously?

Reviewer #3: Yes

Reviewer #4: I Don't Know

Reviewer #5: N/A

Reviewer #6: I Don't Know

Reviewer #7: N/A

Reviewer #8: No

Reviewer #9: Yes

Reviewer #10: I Don't Know

4. Have the authors made all data underlying the findings in their manuscript fully available?

Reviewer #3: Yes

Reviewer #4: No

Reviewer #5: No

Reviewer #6: Yes

Reviewer #7: Yes

Reviewer #8: No

Reviewer #9: Yes

Reviewer #10: Yes

5. Is the manuscript presented in an intelligible fashion and written in standard English?

Reviewer #3: Yes

Reviewer #4: Yes

Reviewer #5: Yes

Reviewer #6: Yes

Reviewer #7: Yes

Reviewer #8: Yes

Reviewer #9: Yes

Reviewer #10: Yes

6. Review Comments to the Author

Reviewer #3: Comments raised by the reviewers has been addressed satisfactorily by the authors, the manuscript is technically and scientifically sound, the content can contribute immensely to the field of antimicrobial resistance.

Response: Thanks so much for reading the manuscript and sharing the encouraging observation. 

Reviewer #4: 

Comment: The research article entitled “An Application of COM-B Model to Explore Factors Influencing Veterinarians’ Antimicrobial Prescription Behaviors: Findings from a Qualitative Study in Bangladesh” has been reviewed. In the study, Shano and co-workers applied the Capability, Opportunity and Motivation for Behaviour (COM-B) model to comprehensively study the behavioural change models to evaluate antimicrobial prescriptive interventions among registered veterinarians in Bangladesh. The study mainly involved one-on-one interviews with 32 participants from the aforementioned professionals. Positively, it could have impacted the policy makers on necessary regulations on the prescription behaviours of antimicrobial prescriptions by veterinarians in Bangladesh and some other nations affected by the knowledge covered. However, most of the highlighted outcomes are ideally expected to have been undertaken by the professionals as important parts of curriculum during the course of training. If so, then what is the significance of this study? Authors need to strongly indicate that the new findings are not already existing knowledge in the course of training professional veterinarians that prescribe antimicrobials or channel their efforts towards the implementation of the known knowledge/policy. Based on this and other comments enlisted below, I recommend a major revision before the manuscript can be accepted for publication in this title.

Response: Thank you for your comment. However, we respectfully disagree with your interpretation of our paper and its merit for several reasons.

Firstly, the veterinary curriculum does not typically include the COM-B model to enhance knowledge. The COM-B model is a behavioral framework applicable to a wide range of health behaviors, beyond just knowledge improvement.

Secondly, our study goes beyond assessing psychological capability, knowledge, or training-related factors. We also examine critical elements such as physical opportunity (e.g., access to testing facilities, environmental context, structural barriers), social opportunity (e.g., social influences, peer pressure, norms), and factors related to negative reflective motivation. Notably, even with substantial knowledge and training on AMR, these additional factors limit participants' ability to prescribe antimicrobials appropriately.

Thirdly, our recommendations extend beyond training and education. We advocate for a comprehensive strategy that includes enhancing physical opportunities and engaging stakeholders.

Finally, our study is one of the few to use the COM-B model as a conceptual framework in low- and middle-income countries. Therefore, the findings of this study are a novel contribution to the existing literature. 

However, as detailed below, we have addressed your remaining comments.

Comment 1: 1. Understandably, the cause of antimicrobial resistance extends beyond an excessive usage but also importantly entails misuse of antimicrobials and several other factors including poor hygiene, lack of sanitation, contamination of food, drink and environments, inadequate preventive and control measures etc. How have the authors considered the influence of these other factors on the prescription behaviour of veterinarians? In addition, evolvement of resistance mechanisms towards the existing medications oftentimes constitute an increase in antimicrobials prescription. Authors need to also consider this in their recommendations.

Response: Thank you for your comment. The primary focus of our study was to understand the prescribing behavior of veterinarians and the factors that influence their decisions to prescribe antimicrobials. While the factors you mentioned are indeed critical in studying AMR, including all of them in a single study would have added considerable complexity and required a different study design. Furthermore, many of these factors have already been explored in the Bangladeshi context by other researchers.

However, our participants did highlight several relevant issues. Based on their responses, we reported factors such as farm management, infection prevention and control, hygiene practices, and the impact of climate change on antimicrobial prescribing. Therefore, our recommendations include strategies aimed at enhancing these specific areas.

Comment: 2. Were the COM-B practices not already in the curriculum or courses of studies undertaken by the formal professionals during their trainings? If yes, what is the significance of this present study? Perhaps the results from this study could discourage. 

Response: No. COM-B is a behavioral model that can be applied in identifying behavioral factors for any health behaviors. Please see the paper about COM-B model for details: https://implementationscience.biomedcentral.com/articles/10.1186/1748-5908-6-42

Comment: 3. Authors are advised to minimize the usage of personalized statements and instead, adopt reporting formats. E.g. use “the study was conducted by…” instead of “we conducted the study by…”

Response: Edited as suggested. Please check page 10, line, 184-185 in the revised version. 

Comment: 4. What influenced the significant gender bias in authors’ criteria for participant selections and how could these not affect the fairness in the outcomes?

Response: Thank you for your comment. Firstly, women's participation in veterinary education is significantly lower than men's, due to various social and cultural factors. As a result, we observed that only a small number of women are registered in veterinary practice, and their representation in the field is very limited. During our recruitment process, we made efforts to include as many women veterinarians as possible. However, we were unable to increase their numbers due to structural and systemic barriers beyond our control.

Comment: 5. In section 3.2, the authors have mentioned bachelor degree holders among the recruited participants. However, this class is missing in the table of participant’s characteristics (Table 1). Kindly clarify.

Response: Thanks for your comment. This category includes with DVM (Doctor of Veterinary Medicine). We have included this in the parenthesis in page 13, line 231, in the revised version. 

Comment: 6. I strongly advise that the knowledge depths of the selected participants should also be considered to validate the results from this study. For instance, if a farmer intends to sell his chickens on 30th and they developed microbial infection on 27th as mentioned by a participant, what should be done by the farmer for his business security and human safety?

Response: Thanks for your comment. The example given by the participant was about their knowledge about describing AMR and its development. We are not validating their knowledge, but a presentation how they explained AMR. This will ultimately reflect on the need of accurate knowledge through either training or workshop as part of antimicrobial stewardship initiative. 

Comment: 7. Were the quantitative data from this study analyzed statistically? Kindly indicate the location if available.

Response: This is a qualitative study, not quantitative. Hence, we do not have any quantitative data or statistical analysis. 

Comment: 8. The English language of this submission requires a thorough recheck by an expert. Authors should consider this.

Response: This manuscript was read by a native individual who independently checked grammar and English. Additionally, one of the authors (senior author) is a native English speaker who also checked and improved English during the previous rounds of resubmission. 

Reviewer #5: (No Response)

Reviewer #6: The previous comments have been addressed. However, in responses, Line 281 it is mentioned that ciprofloxacin is narrow spectrum antibiotic. Kindly check the transcript again or otherwise this still shows some knowledge gaps as ciprofloxacin is broad spectrum.

Response: Thanks so much for your concern. We have checked the audio recording and transcript of that interview. We have now corrected with ‘broad’ in the revised manuscript. Please check page 16, line number 281 in the revised version. 

Reviewer #7: Both reviewers are well explained each of the statement. Its an interesting survey work. But one things, Its not a clear idea how antibiotic resistance created human body after eating poultry products?

Response: thanks for your comment. The relative explanations are participants reasoning on AMR. As we mentioned in the manuscript, participants used a variety of explanations, and identified several pathways including the respecting the withdrawal periods. In the case of chicken, for example one quotation (page 15, lines 255-260) specifically indicated, if the withdrawal period is not respected, administering antibiotics to poultry shortly before slaughter can lead to antibiotic residues in the meat. When humans consume this meat, it contributes to the development of antibiotic-resistant bacteria in their bodies. 

Reviewer #8: Dear authors, I have the following concerns

Comment 1. The sample size (32) is too small to draw strong evidence to fulfil the objectives. according to the Bangladesh Veterinary Council, the country has more than eight thousand veterinarians, then how these 32 will present the other. This was also expressed under the limitation. However, it is too small to fulfil objectives.

Response: Thank you for your comment regarding the sample size. We respectfully disagree that a sample size of 32 interviews is too small for our study. We believe the sample size was appropriate based on the following qualitative research principles. 

Firstly, saturation achieved: In qualitative research, the goal is not to generalize to the entire population but to achieve saturation, where no new themes or insights emerge from additional data collection. In our study, we reached saturation after 32 interviews, indicating that the sample size was sufficient to capture the breadth and depth of relevant themes.

Secondly, diverse sampling for rich data: We ensured a diverse and purposive sampling strategy by including participants from various backgrounds, including different genders, years of practice (experience), geographic locations (urban vs. rural), nature of affiliations (government vs. non-government), and educational qualifications (bachelor’s, master’s, or higher degrees). This diversity in sampling enhances the transferability of our findings by capturing a wide range of perspectives, making our results more robust.

Thirdly, credibility and reliability of findings: Our methodological approach adhered to principles of credibility and reliability in qualitative research. We used a well-structured interview guide, conducted interviews until no new information was gained (saturation), and employed strategies such as member checking and triangulation to ensure the accuracy and consistency of our findings.

Fourthly, qualitative research purpose: The purpose of qualitative research is to explore and understand complex phenomena in-depth rather than to generalize to a larger population. Our focus was on understanding the nuances of veterinarians’ prescribing behaviors and the factors influencing them. Given this exploratory nature, a smaller, carefully selected sample is not only adequate but also consistent with qualitative research principles.

Finally, acknowledge and address limitations: While we acknowledge the limitation regarding the sample size in relation to the larger population of veterinarians, we addressed this in our discussion. The intent of our study was not to provide statistical generalizations but to gain in-depth insights into prescribing behaviors, which are best captured through qualitative methods.

Comment 2. Line 153, remove phrases like ----please see, put only something in the for further reference. 

Response: Removed as suggested. Please check line number 153 in the revised version. 

Comment 3. Line 255, instead of saying “some of the participants” why don't you write the number and/or of participants who answered that question. This is true throughout the document including the abstract

Response: Thank you for your comment and suggestion. However, we would like to clarify that the primary purpose of qualitative research is not to quantify the number of participants who expressed a particular view, but rather to provide in-depth insights, thoughts, and themes that emerge from the data. Using phrases like "some of the participants" is intentional, as it aligns with qualitative research principles, which focus on understanding the richness and

---

## [Editor Report · Decision Letter 3]

22 Nov 2024

An Application of COM-B Model to Explore Factors Influencing Veterinarians’ Antimicrobial Prescription Behaviors: Findings from a Qualitative Study in Bangladesh.

PONE-D-23-31043R3

Dear Dr. Kalam,

We’re pleased to inform you that your manuscript has been judged scientifically suitable for publication and will be formally accepted for publication once it meets all outstanding technical requirements.

Kind regards,

Muhammad Ahmad

Academic Editor

PLOS ONE
---

## [Editor Report · Acceptance letter]

1 Dec 2024

PONE-D-23-31043R3 

PLOS ONE

Dear Dr. Kalam, 

I'm pleased to inform you that your manuscript has been deemed suitable for publication in PLOS ONE. Congratulations! Your manuscript is now being handed over to our production team.

Kind regards, 

on behalf of

Mr. Muhammad Ahmad 

Academic Editor

PLOS ONE